# Co-infections and risk factors of *Toxoplasma gondii* infection among pregnant women in Ghana: A facility-based cross-sectional study

**Ebenezer Assoah**[1]*, **Denis Dekugmen Yar**[2], **Papa Kofi Amissah-Reynolds**[1], **Gadafi Iddrisu Balali**[3,4], **Rockson Addy**[2,5], **Joshua Kpieonuma Zineyele**[1]

**1** Department of Biological Sciences Education, Akenten Appiah-Menka University of Skills Training and Entrepreneurial Development (AAMUSTED), Asante Mampong Campus, Mampong, Ghana, **2** Department of Public Health Education, Akenten Appiah-Menka University of Skills Training and Entrepreneurial Development (AAMUSTED), Asante Mampong Campus, Mampong, Ghana, **3** Department of Theoretical and Applied Biology, Kwame Nkrumah University of Science and Technology, Kumasi, Ghana, **4** Department of Science Education, Seventh–Day Adventist College of Education, Agona-Ashanti, Ghana, **5** Department of Science and Information Communication Technology, Effiduase Senior High Technical School, Effiduase, Ashanti Region, Ghana

* assoahebenezer16@gmail.com

## Abstract

This study assessed the prevalence of co-infections (human immunodeficiency virus, hepatitis B, and syphilis) and associated risk factors for *Toxoplasma gondii* infection among pregnant women in Mampong Municipality, Ghana. A cross-sectional design was used to recruit 201 pregnant women from six health facilities conveniently. Participants' socio-demographics, clinical and environmental data were collected using a structured questionnaire. Using 2 ml of blood, *T. gondii* seroprevalence was determined by the TOXO IgG/IgM Rapid Test Cassette. Data was analyzed using descriptive and logistic regression analysis with SPSS version 27 to determine the prevalence and associations of *T. gondii* infection with other variables, respectively. The seroprevalence of *T. gondii* was 49.75%, of which 40.30%, 2.49%, and 6.97% tested positive for IgG, IgM, and IgG/IgM, respectively. Co-infection of toxoplasmosis with viral hepatitis B, human immunodeficiency virus (HIV), and syphilis rates were 15%, 1%, and 4%, respectively and were not risk factors for *T. gondii* transmission. Educational level and residential status were associated with toxoplasmosis [p < 0.05]. Participants with higher education had a reduced risk of *T. gondii* infections compared to a lower level of education [AOR = 0.39 (0.13, 0.99) p = 0.049]. Similarly, the risk of *T. gondii* infection was significantly lower among individuals residing in peri-urban [AOR = 0.13 (0.02–0.70), p = 0.02] and urban areas [AOR = 0.10 (0.02–0.78), p = 0.03] compared to those in rural areas. Backyard animals with extensive and semi-intensive systems, without veterinary care, and contact with animal droppings and water sources were significant risk factors for *T. gondii* infection [p < 0.05]. Miscarriage was associated with *T. gondii* infection [p < 0.05]. The burden of *T. gondii*

**Data availability statement:** All relevant data are within the manuscript.

**Funding:** The author(s) received no specific funding for this work.

**Competing interests:** The authors have declared that no competing interests exist.

infection was high among the study population, posing a risk of mother-to-child transmission. Key risk factors included low education, rural residence, backyard animal exposure, poor hygiene, and unsafe water sources. Toxoplasmosis was associated with miscarriage; thus, integrating it into routine antenatal screening could improve pregnancy outcomes. Health promotion interventions such as education on zoonotic risks, improved sanitation, safe water practices, and veterinary care for domestic animals are recommended to reduce infection risk among pregnant women.

## Introduction

Globally, parasitic zoonoses are on the rise, affecting more than 4 billion people, with over 200,000 deaths recorded annually [1,2]. In low-income countries, neglected parasitic diseases such as ascariasis, amebiasis, giardiasis, toxoplasmosis, etc., continue to cause significant public health challenges, with enormous morbidity and mortality [3]. *Toxoplasma gondii* is known to infect vertebrate animals and humans [4,5], with 25.7% and 33.8% reported among the general human population and pregnant women, respectively [6,7]. The most significant burden is in developing countries, linked to poor sanitation and a lack of quality health services [7–10]. Children and pregnant women are generally more vulnerable to *T. gondii* infection due to their weaker immunity [11]. Pregnant women infected with *T. gondii* in the first trimester can invade the placenta, causing inflammation and damage to arteries and veins that may block the flow of blood and nutrient exchange between the mother and fetus, resulting in spontaneous abortion [12]. Meanwhile, *T. gondii* infection in the second and third trimesters may cause congenital toxoplasmosis, leading to mental retardation, auditory defects, and chorioretinitis [13,14].

In Ghana, policies for screening expectant mothers exist for a range of maternal infections with potential for vertical transmission such as viral hepatitis, HIV/AIDS, syphilis and malaria, to improve pregnancy outcomes and neonatal health [15,16]. The prevalence of toxoplasmosis among pregnant women in Ghana is among the highest in the world (81–90%), with possible dire consequences on pregnancy outcomes [7,17]. Nevertheless, there is no policy for *T. gondii* screening during pregnancy in the country. However, in Europe (Austria, France, and Slovenia), mandatory national policies for prenatal serological screening and treatment for *T. gondii* infection have significantly reduced congenital toxoplasmosis [18]. Early detection and administration of anti-*Toxoplasma* prophylaxis during pregnancy prevent congenital transmission and reduce sequelae in neonates [18].

Over the past two decades, significant progress has been made in maternal health in Ghana. Notwithstanding, maternal and neonatal morbidity and mortality are still on the increase, attributable to failure to screen and treat these diseases, including toxoplasmosis [19]. Several studies have assessed the prevalence of *T. gondii* infection among expectant mothers, with limited focus on its comorbidity with maternal infections and its transmission dynamics in Ghana [17,20]. Meanwhile, data on miscarriages among pregnant women over the last decade in the Mampong Municipality

is on the rise [21]. However, the proxy determinants of the high miscarriage rates are uncertain and could be linked to toxoplasmosis and co-infections [22].

There is limited literature on the causal relationship between transmission dynamics and co-infection of *T. gondii* with HIV, viral hepatitis B, and syphilis in pregnancy in the Municipality. Hence, this study investigated the prevalence of co-infections (HIV, hepatitis B, and syphilis) and associated risk factors for *T. gondii* infection among pregnant women in Mampong Municipality, Ghana.

## 2. Materials and methods

### Study design

This was a facility-based cross-sectional study that assessed the burden and concurrence of *Toxoplasma gondii* infection with HBV, HIV, and syphilis among pregnant women attending antenatal care (ANC) services in health facilities. The study also examined the risks of *T. gondii* transmission in the Mampong Municipality. Following the approval of the study on 11th August 2023, participant recruitment and data collection commenced on 15th August 2023 and concluded on 15th May 2024.

### Study area

**This study was conducted in the Asante Mampong Municipality of the Ashanti Region of Ghana**. The study was conducted in six health facilities in five communities, with Mampong as its capital town, Dadease, Krobo, Asaam, and Kofiase. The municipality has a total population of 116,632, with females slightly more than males [23]. The municipality has an average temperature of 28°C with a relative humidity of 63% [23]. The main economic activity in the municipality is agricultural activities, including animal farming, engaging around 67.30% of the workforce [23,24]. Most households in the Municipality own domesticated animals, such as sheep, goats, poultry, cattle, cats, and dogs.

### Study site

Data for the study were collected from the various health facilities with maternal and laboratory departments in the municipality. Mampong Government Hospital-Maternity Home, Calvary Health Center, Sister Phillipah Maternity Home and Clinic, Krobo-Dadease Health Center, Asaam Health Center, and Kofiase Health Center were used for the study, which has most pregnant women attending ANC. These facilities were purposefully chosen based on their distinct characteristics that would ensure balanced urban and rural data to reduce bias. The facilities were categorized into government and private hospitals, health centres, and clinics. This approach consequently yielded a substantial dataset suitable for making inferences.

### Study population and sample size

Pregnant women of reproductive age (15–49 years) visiting the six selected health facilities for prenatal care and related services in the Mampong municipality were included in the study. Participants who had undergone laboratory and ultrasonography tests confirming their pregnancy and had lived in the study area for at least six months were recruited for the study. A sample size of 201 was estimated based on the toxoplasmosis prevalence rate of 92%, using Cochran's formula as previously described [25,26].

### Data collection tools and techniques

A structured questionnaire consisted of four sections, including socio-demographic characteristics such as age, religion, education, and residential status; obstetric variables such as trimester and gravidity; laboratory test results such as HBV, HIV, and syphilis; and environmental variables such as water sources and the presence and confinement levels of

backyard farm animals, and pets. The study participants attending antenatal care were conveniently sampled at all the selected health facilities.

## Data collection procedure

After obtaining informed consent, face-to-face interviews and observational methods were used to gather data from participants using a structured questionnaire. Information on socio-demographic characteristics, environmental exposures, and obstetric history was collected. In addition, routine screening test results recorded in the participants' antenatal care (ANC) record books were reviewed and extracted.

All 201 participants were screened for *T. gondii* using the TOXO IgG/IgM Rapid Test Cassette. However, only 178, 156, and 163 participants had been tested for hepatitis B virus (HBV), HIV, and syphilis, respectively, at the time of data collection. These tests are part of the routine ANC screening; however, some participants reported that they were unable to complete all required tests due to financial constraints. This limitation was considered during data analysis, and missing values were handled appropriately.

## Blood sample collection and laboratory methods

A 2 ml of each participant's blood was collected using venipuncture and into EDTA tubes. This was centrifuged at 1780 x g for 10 minutes at $4^0$C to separate plasma for subsequent serological analysis. The plasma was promptly assayed for anti-*T. gondii* IgG and IgM antibodies using the immunochromatography Test (ICT) technique. The specific ICT used was the TOXO IgG/IgM One-Step Rapid Test Cassette (WB/S/P) produced by Evancare Medical (Nantong) Co., Ltd. (China). This assay employs a lateral flow chromatographic immunoassay for the simultaneous detection and differentiation of IgG and IgM anti-*T. gondii* in human serum, plasma, or whole blood. Testing procedures were strictly followed according to the manufacturer's instructions [27].

## Statistical analysis

Data were entered into Microsoft Excel (version 2016) and exported into IBM SPSS version 27.0 for statistical analysis. All variables used in the analysis were categorical in nature. Some variables, such as *T. gondii*, HBV, HIV, syphilis test results and other risk factors were binary (e.g., positive/negative), while others like age group, education level, and other variables had more than two levels.

Descriptive statistics were applied to determine the frequencies and percentages of *T. gondii*, HBV, HIV, syphilis, and other socio-demographic and environmental characteristics. The co-infection rate was calculated by dividing the number of individuals positive for both *T. gondii* and any of the ANC-related infections (HBV, HIV, or syphilis) by the total number of *T. gondii*-positive individuals, and expressed as a percentage:

$$Co-Infection\ Rate\ (\%) = (Number\ of\ individuals\ positive\ for\ both$$
$$T.\ gondii\ and\ ANC\ infections) / Total\ number\ of\ T.\ gondii-positive\ individuals) \times 100\%.$$

Chi-square test of independence and binary logistic regression were used to assess the association between *T. gondii* infection and other variables. Prior to conducting logistic regression, model fitness was evaluated using the Hosmer-Lemeshow goodness-of-fit test ($p > 0.05$ indicating good fit). Additionally, the Nagelkerke $R^2$ statistic and the model's classification accuracy were examined to assess the model's explanatory power and predictive ability.

For the multivariate logistic regression analysis, all variables with a p-value less than 0.2 in the univariate analysis were considered for inclusion. Variables with more than two levels were included as categorical factors using dummy coding in SPSS. Variables of known epidemiological importance were also retained in the final model regardless of statistical significance. A p-value < 0.05 and a 95% confidence interval were used to determine statistical significance.

### Ethical review and clearance

The study received ethical approval from the Committee on Human Research, Publications, and Ethics (CHRPE) of the Kwame Nkrumah University of Science and Technology, under reference number CHRPE/AP/717/23. Following ethical clearance, official permissions were obtained from the Ashanti Regional and Asante Mampong Municipal Health Directorates, as well as from the respective health facilities where the research was conducted.

Before data collection began, each participant was given a detailed verbal explanation of the study's objectives, procedures, potential risks, and benefits. They were also informed of their right to voluntary participation and their freedom to withdraw from the study at any time without any consequences. After this briefing, participants who agreed to take part signed a written informed consent form, confirming their voluntary participation.

## 3. Results

### Seroprevalence of *T. gondii*

The prevalence of *T. gondii* infection among the participants was 49.75% (100/201), of which 40.30% (81/201), 2.49% (5/201), and 6.97% (14/201) tested positive for IgG, IgM, and IgG/IgM, respectively (Fig 1).

### Prevalence of Hepatitis B, HIV and syphilis among pregnant women in Asante Mampong Municipal of Ghana

In Table 1, the prevalence of HBV, HIV, and syphilis among pregnant women screened during ANC was 14.61%, 0.61% and 3.07%, respectively. There were variations in the sample sizes occasioned by the non-availability of routine screening tests at the ANC clinics for these routine ANC tests.

### Toxoplasmosis concurrence with HBV, HIV, and syphilis screened among participants

Table 2 shows the prevalence of co-infection of toxoplasmosis with HBV, HIV, and syphilis were 15%, 1%, and 4%, respectively. *T. gondii* infection was not associated with the HBV, HIV, and Syphilis screened at ANC (**P-value > 0.05).**

### Association of *T. gondii* infection with sociodemographic characteristics of participants

In Table 3, the level of education and residential area were significantly associated with *T. gondii* infection ($x^2 = 6.56$, p = 0.04) and ($x^2 = 10.97$, p = 0.004), respectively. Participants with tertiary education were less likely to be infected with *T. gondii* infections than those with lower levels of education [AOR = 0.39 (0.13–0.99) p = 0.049]. Expectant mothers who reside in the peri-urban and urban areas were less likely to be infected with *T. gondii* as compared with rural residents

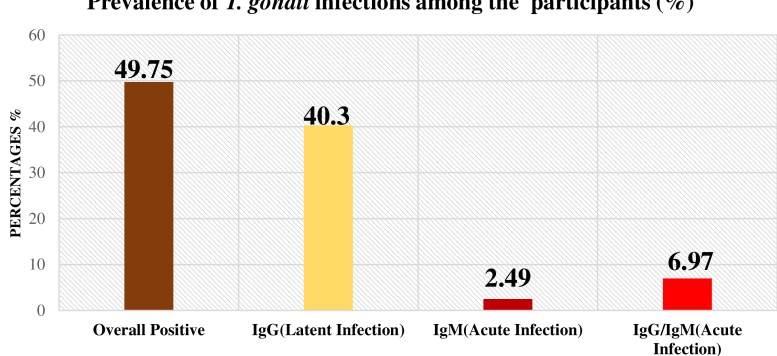

**Fig 1. Seroprevalence of *T. gondii* infection in the study participants.**

**Table 1. Prevalence of Hepatitis B, HIV and syphilis among participants.**

| Infections Screened | Frequency (N) | Percentage % |
|---|---|---|
| **Hepatitis B Test (N = 178)** | | |
| Positive | 26 | 14.61 |
| Negative | 152 | 85.39 |
| **HIV Test (N = 165)** | | |
| Positive | 1 | 0.61 |
| Negative | 164 | 99.39 |
| **Syphilis test (N = 163)** | | |
| Positive | 5 | 3.07 |
| Negative | 158 | 96.93 |

**Data Source: Laboratory records, 2023**

**Note: N = sample size for the individual test.**

[AOR = 0.13 (0.02–0.70) p = 0.02; AOR = 0.10 (0.02–0.78) p = 0.03]. Age, religion, gestational period, and gravida were not associated with toxoplasmosis (**P-value > 0.05**).

### Risks factors of *T. gondii* infection among pregnant women

**Table 4** shows that participants with backyard animals, with the extensive and semi-intensive system, without veterinary care, and contact with animal faeces had significant odds of being infected with *T. gondii* [AOR = 3.90 (1.60–9.48) p = 0.003], [AOR = 31.7 (20.90–8921.3) p < 0.001]; AOR = 5.2 (3.37–367.70) p = 0.003], [AOR = 2.90 (1.10–7.46) p = 0.031], and [AOR = 3.90 (1.81–8.35) p = 0.001], respectively, whereas the use of disposable gloves to hand animals' faeces significantly reduced the odds [AOR = 0.05 (0.002–0.91) p = 0.043]. Table 4 further shows that participants using river, stream, or well water had 2.91 times the odds of being infected with *T. gondii* infections [AOR = 2.91 (1.07–7.92) p = 0.037], whereas participants who treated their water before drinking had less risk of the infection [AOR = 0.26 (0.10–0.70) p = 0.008]. There was no association between eating undercooked animal meat and drinking unpasteurized milk with *T. gondii* infections (**P-value > 0.05**). It should be noted that response totals for certain variables may be less than 100 due to skip logic in the questionnaire design, where only applicable participants responded to follow-up questions.

### Association between toxoplasmosis and obstetric factors

**Table 5** shows that pregnant women with *T. gondii* infection were associated with miscarriage ($x^2 = 6.1$, p = 0.014). Participants who had ever had a miscarriage were 2.93 times more likely to be infected with *T. gondii* [AOR = 2.93 (1.20–7.04),

**Table 2. Toxoplasmosis concurrence with HBV, HIV, and syphilis screened at ANC.**

| Infections | Toxoplasmosis Positive, N = 100 (Co-infection Rate %) | $X^2$ (P-Value) |
|---|---|---|
| **HBV Positive** | **15 (15.0)** | 0.44 **(0.507)** |
| **HIV Positive** | **1 (1.0)** | 1.04 **(0.307)** |
| **Syphilis Positive** | **4 (4.0)** | 1.67 **(0.196)** |

**Data Source: Laboratory records, 2023**

**Note:** $Co-Infection\ (Number\ of\ positives\ for\ ANC\ diseases / Total\ number\ of\ positives\ for\ T.\ gondii)\ \times\ 100\%$.

$X^2$ = Chi-square test of independence

**Table 3. Association of toxoplasmosis with Participants' Socio-demographic characteristics.**

| Demographics | *T. gondii* Positive (%) | χ² (P- v) | COR (95%CI) p-v | AOR (95%CI) p-v |
|---|---|---|---|---|
| **Age range** | | | | |
| 15–19 | 18 (58.1) | 1.71 (0.43) | Ref | Ref |
| 20–30 | 49 (51.0) | | 0.68 (0.25, 1.34) 0.21 | 0.68 (0.26, 1.74)0.42 |
| 31–49 | 33 (44.6) | | 0.77 (0.42, 1.42) 0.41 | 0.95 (0.47, 1.89)0.88 |
| **Religion** | | | | |
| Christians | 89 (51.7) | 1.89 (0.17) | Ref | Ref |
| Muslims | 11 (37.9) | | 1.755 (0.78, 3.94) 0.17 | 2.1 (0.86, 5.26) 0.10 |
| **Education** | | | | |
| None | 10 (55.6) | **6.56 (0.04)** | Ref | Ref |
| Primary | 49 (47.6) | | 0.48 (0.16, 1.48) 0.20 | 0.39 (0.11, 1.30) 0.12 |
| SHS | 26 (65) | | 0.66(0.31, 1.40) 0.28 | 0.72 (0.33, 1.60) 0.42 |
| Tertiary | 15 (37.5) | | 0.32(0.13, 0.80) **0.015** | **0.39**(0.13, 0.99) **0.049** |
| **Residential** | | | | |
| Rural | 40 (67.8) | **10.97(0.04)** | Ref | Ref |
| Peri-urban | 22 (44) | | 0.37 (0.17, 0.82) **0.013** | 0.13 (0.02, 0.7) **0.02** |
| Urban | 38 (41.3) | | 0.33 (0.17, 0.66) **0.02** | 0.10 (0.02, 0.78) **0.03** |
| **Trimester** | | | | |
| First | 13 (52) | 0.07 (0.97) | Ref | Ref |
| Second | 34 (50) | | 1.12 (0.47, 2.69) 0.78 | 1.3 (0.51, 3.30) 0.58 |
| Third | 53 (49.1) | | 1.04 (0.57, 1.90) 0.91 | 1.1 (0.58, 2.12) 0.75 |
| **Gravidity** | | | | |
| Gravida I | 25 (53.2) | 1.14 (0.57) | Ref | Ref |
| Gravida II | 25 (43.9) | | 1.46 (0.67, 3.16) 0.344 | 1.37 (0.54, 3.48) 0.50 |
| >Gravida II | 50 (51.5) | | 1.07 (0.53, 2.14) 0.85 | 0.99 (0.38, 2.62) 0.99 |

**Data Source: Laboratory records, 2023**

**Note:** *P-v = P-Value; Ref = reference; CI = confidence interval.*

p = 0.016]. It should be noted that response totals for certain variables may be less than 100 due to skip logic in the questionnaire design, where only applicable participants responded to follow-up questions.

## 4. Discussion

### Seroprevalence of *T. gondii*

The seroprevalence of *T. gondii* infection in this study was 49.75% among pregnant women. This finding is consistent with previous studies conducted in Ghana, which reported seroprevalence rates ranging from 50% to 56% among pregnant women attending health facilities [20,28,29]. However, some studies conducted in other regions of Ghana have reported significantly higher prevalence rates, exceeding 76% [30–32]. These within-country variations may be attributed to differences in geographic settings, urban–rural residency, levels of sanitation, pet ownership, access to veterinary services, and socioeconomic conditions. The relatively lower prevalence in the current study could reflect better hygienic practices or improved awareness in the study area compared to regions with higher reported prevalence.

In contrast, when comparing with studies from other countries, the seroprevalence reported in this study was higher than findings from Ethiopia (35.6%) in different settings [33], but relatively lower than rates reported in Western Romania (55.8%) and Brazil (71%) [34,35]. These international disparities may result from differences in climate, dietary habits (such as consumption of undercooked meat), environmental exposures, and public health infrastructure. Additionally,

**Table 4. Association between environmental factors and *T. gondii* infection among the study population.**

| Risk factors | *T. gondii* Positive = 100 (%) | χ² (P-value) | COR (95%CI) p-v | AOR (95%CI) p-v |
|---|---|---|---|---|
| Animal in household | | | | |
| Yes | 92 (58.6) | 22.5 (0.001) | 6.4 (2.78, 14.60) **0.001** | **3.9** (1.60, 9.48) **0.003** |
| No | 8 (18.2) | | Ref | Ref |
| Type of animals | | | | |
| Pet (cat, dog, etc.) | 51 (56.7) | 0.5 (0.779) | 0.75 (0.29, 1.96) 0.55 | 0.96 (0.33, 2.81) 0.93 |
| Poultry/birds | 27 (61.4) | | 0.91 (0.32, 2.62) 0.86 | 1.13 (0.35, 3.66) 0.84 |
| Ruminant | 14 (63.6) | | Ref | Ref |
| Level of confinement | | | | |
| Extensive system | 71 (94.7) | 92.1 (0.001) | 16.9(42.6, 653.6)**0.001** | **31.7**(20.9, 8921.3) **0.001** |
| Semi-intensive | 17 (56.7) | | 4.3 (3.81, 39.65) **0.001** | **5.2** (3.37, 367.7) **0.003** |
| Intensive system | 5 (9.6) | | Ref | Ref |
| Contact with animal droppings | | | | |
| Yes | 54 (66.7) | 11.9 (**0.001**) | 3.4 (1.67, 6.78) **0.001** | **3.9** (1.81, 8.35) **0.001** |
| No | 22 (37.3) | | Ref | Ref |
| Veterinary care | | | | |
| Yes | 22 (50) | 6.37 (**0.041**) | 1.3 (0.47, 3.59) 0.61 | 1.4 (0.45, 4.10) 0.51 |
| No | 60 (67.4) | | 2.7 (1.06, 6.86) **0.038** | **2.9** (1.10, 7.46) **0.031** |
| Do not know | 10 (43.5) | | Ref | Ref |
| Use gloves to handle animal faeces. | | | | |
| Yes | 2 (25) | 14.7 (**0.001**) | 0.07 (0.01, 0.38) **0.002** | **0.05** (0.002, 0.91) **0.043** |
| No | 90 (82.6) | | Ref | Ref |
| Eating of animal meat | | | | |
| Well cooked | 71 (48.3) | 0.46 (0.497) | 0.81 (0.43, 1.51) 0.50 | 0.80 (0.43, 1.50) 0.49 |
| Undercooked or raw | 29 (53.7) | | Ref | |
| Take unpasteurized milk | | | | |
| No | 74 (49.3) | 0.04 (0.839) | 0.94 (0.49, 1.77) 0.84 | 0.93 (0.50, 1.76) 0.83 |
| Yes | 26 (51) | | Ref | |
| Primary source of drinking water | | | | |
| Pipe | 45 (39.5) | 14.9 (0.002) | 0.61 (0.28, 1.34) 0.22 | 0.60 (0.27, 1.34) 0.21 |
| Borehole | 7 (58.3) | | 1.3 (0.35, 5.01) 0.69 | 2.72 (0.60, 12.23) 0.19 |
| River/stream/well | 31 (73.8) | | 2.7 (1.01, 6.99) **0.049** | 2.91 (1.07, 7.92) **0.037** |
| Bottle/sachet water | 17 (51.5) | | Ref | Ref |
| Treat water before drinking. | | | | |
| Yes | 9 (29) | 6.3 (0.012) | 0.36 (0.16, 0.82) **0.015** | 0.26 (0.10, 0.70) **0.008** |
| No | 91 (53.5) | | Ref | Ref |
| Share water sources with animals. | | | | |
| Yes | 19 (76) | 7.9 (0.005) | 3.7(1.42, 9.74) **0.008** | 2.2 (0.76, 6.40) 0.14 |
| No | 81 (46) | | Ref | |

**(Data Source: Field Data and Laboratory records, 2023)**

**Note:** P-v = P-Value; Ref = reference; CI = confidence interval.

Note: Total number of responses for some variables may be less than 100 due to conditional (skip) questions or non-responses.

**Table 5. Association between toxoplasmosis and obstetric factors.**

| Risk factors | T. gondii Positive (%) | χ² (P-value) | COR (95%CI) p-v | AOR (95%CI) p-v |
|---|---|---|---|---|
| **Child with hearing loss** | | | | |
| Yes | 1 (50) | 0.00 (1.0) | 1.0 (0.06, 16.3) 1.0 | 1.7 (0.10, 29.3) 0.72 |
| No | 70 (50) | | Ref | Ref |
| **Miscarriage history** | | | | |
| Miscarriage in past | 24 (66.7) | **6.1 (0.014)** | 2.6 (1.20, 5.75) **0.016** | 2.93 (1.2, 7.04) **0.016** |
| No miscarriage | 51 (43.2) | | Ref | Ref |
| **Had chorioretinitis** | | | | |
| Yes | 34 (55.7) | 1.26 (0.263) | 1.4 (0.77, 2.58) 0.26 | 1.8 (0.82, 3.93) 0.14 |
| No | 66 (47.1) | | Ref | Ref |

**Data Source: Field Data and Laboratory records, 2023**

Note: Total number of responses for some variables may be less than 100 due to conditional (skip) questions or non-responses.

methodological differences such as serological testing tools, sample sizes, and study designs can influence prevalence rates across countries [36,37].

The current study also found that participants residing in rural areas and those with lower socioeconomic status were more vulnerable to *T. gondii* infection, likely due to poorer sanitation, limited access to clean water, and close contact with animals. Key risk factors included poor management of pets and backyard livestock, lack of veterinary care, and exposure to contaminated water sources, all of which are more common in rural settings [38,39].

Importantly, approximately 10% of participants had acute infection, with about 3% being seronegative (naïve), suggesting a high potential risk of mother-to-child transmission of *T. gondii*, which can have serious implications for fetal development and pregnancy outcomes [20,40,41].

### Prevalence of HBV, syphilis and HIV Screened at ANC among the participants

Ghana, over the years, has formulated policies for screening expectant mothers for viral hepatitis, HIV/AIDS, syphilis, and malaria to improve pregnancy outcomes and neonatal health [15,16]. Nevertheless, this study reported a high prevalence of 14.61%. 3.07% and 0.61% for hepatitis B, syphilis, and HIV infections, respectively, among the participants.

The rate of viral hepatitis B (HBV) infection was much higher than in previous studies in Ghana which reported between 2.4% and 10.6% among pregnant women [42–50]. These disparities in HBV prevalence among pregnant women in Ghana could be attributed to the different settings and cultural and behavioural characteristics of the participants [51]. In 2002, Ghana initiated a national expanded program for hepatitis B immunization [52] in response to the WHO target to eliminate HBV infection by 2030 [53,54]. However, this current rate of HBV infection suggests a low vaccine uptake, and the 2030 target may not be achieved. This current rate presents a high risk for mother-to-child transmission, which is the main route of HBV infection, including sexual and other behavioural characteristics [52,55].

The seroprevalence of syphilis was 3.07% among the participants, well aligned with the 3.7% and 4.1% reported among pregnant women at the Cape Coast Metropolitan Hospital [56] and gold miners in Konongo [57] in Ghana. This rate is, however, higher than 0.4%, previously reported in the general population [58]. Several studies have also reported syphilis among pregnant women: 2.5% in Tanzania [59], 1.9% in Ethiopia [60] and 4.4% in Brazil [61]. These prevalence rates somewhat differ from value reported in our study. The variations in these rates may be associated with several factors, including sexual behaviours, lifestyle factors, availability and access to screening services, and cultural and geographical settings [62].

This current study reported a 0.61% HIV infection rate, much lower than the 1.89%, 1.91%, and 1.66% among the adult population, in the Asante Mampong municipal, Ashanti region and the nation, respectively [63]. The current study result is

at variance with 2.8% reported among pregnant women in the Asante Municipality in an HIV Sentinel survey [58]. The low HIV prevalence in this current study could be due to several causes, including the small number of participants screened, the sample size, and the data collection duration. This study observed a very low screening rate for HIV among the participants, and this can be a drawback for the zero-prevalence expected among mothers.

## Toxoplasmosis concurrence with HBV, HIV, and syphilis screened at ANC

This study reported 15%, 1%, and 4% *T. gondii* concurrence with HBV, HIV, and syphilis, respectively, among the population. Although these co-infections were observed, no statistically significant association was established between *T. gondii* infection and any of HBV, HIV, and syphilis screened [64]. This suggests that the observed co-occurrences may be incidental rather than indicative of synergistic interactions or shared transmission pathways.

Nonetheless, the clinical implications of co-infections, particularly during pregnancy, warrant further attention. For instance, co-infection with HIV has been shown in other studies to impair immune responses, potentially increasing the susceptibility to opportunistic infections such as toxoplasmosis and worsening pregnancy outcomes [65–67]. A study in Tanzania reported that toxoplasmosis co-infection could increase mortality among people living with HIV [68]. Although our study did not find such an association, the low HIV prevalence in our sample may have limited the statistical power to detect this relationship.

Given the potential implications of co-infections on maternal and neonatal health, this finding reinforces the need for independent and comprehensive screening protocols for *T. gondii*, HBV, HIV, and syphilis as part of routine antenatal care. Preventive strategies should be tailored to address each condition separately while also considering their possible interactions in immunocompromised individuals.

It is important to note that this study was conducted exclusively within the Asante Mampong Municipality, which may limit the generalizability of the findings to other regions of Ghana or populations with differing socio-economic or epidemiological profiles. Future studies with broader geographic coverage and larger sample sizes are recommended to explore regional variations and more comprehensively assess the dynamics of co-infections in pregnancy.

## Association of *T. gondii* infection with socio-demographic characteristics of participants

Although the prevalence of *T. gondii* decreases with increasing age in consonance, as in previous studies in Italy [69] and Pakistan [70], there was no association. However, this finding contradicts earlier reports in Nigeria [71], the USA [72], and the UK [73], which associated increasing age with *T. gondii* infection. The sharp difference could be attributable to our study's generally younger population of women in their reproductive age.

The findings in this study demonstrated that respondents' educational status and residential area were associated with *T. gondii* infection, as earlier reported [74]. Participants with primary or informal education had an increased risk of *T. gondii* infection in consonance with a report from Ethiopia [75]. Rural dwellers had an increased risk of infection than urban residents, probably due to the inadequate and deplorable social amenities and environmental factors [76]. The plausible reasons for the increased risk of *T. gondii* infection among participants with low education and rural residents may be due to their generally low socio-economic status, limited knowledge of the disease transmission, exposure to backyard animals and pets, and limited access to health facilities, and screening and treatment [77–79].

## Transmission dynamics of *T. gondii* infection among pregnant women

This study has shown that the presence of backyard farm animals in households, their levels of confinement, exposure to faecal droppings, and veterinary care services were significant risk factors for *T. gondii* transmission. Cat is the primary host of *T. gondii,* with several other domestic animals as secondary and mechanical hosts [80], and their interaction with humans can transmit the parasites through their fur, saliva, or faeces [81]. This assertion is affirmed by previous studies that implicated the interaction of domesticated cats and dogs as a risk of *T. gondii* transmission to humans [82,83]. This study also revealed that

backyard animal husbandry practices increase the risk of *T. gondii* infection. The free-range backyard animal-rearing practice is more associated with rural households with little or no veterinary care, increasing the risk of *T. gondii* transmission. These animals could contaminate the environment with their faeces, which contain parasites and pose a risk to humans. In this study, contact with animal faeces increased the risk of *T. gondii* infection four-fold. This implies that droppings and litter from animals in the environment and poor hygiene practices can facilitate the transmission of *T. gondii* to humans [84]. However, using personal protective equipment to handle animals' faeces highly reduced the risks of *T. gondii* infection.

In this study, the consumption of animal products was not associated with *T. gondi*i infection, contrary to an earlier report [64] that linked eating meat and other animal products to the infection [85]. This study has shown that there is a three-fold increased risk of *T. gondii* infection in drinking water from rivers, streams, or wells, akin to a study in Brazil [86]. Livestock often share rivers and streams with humans and are most likely contaminated with faecal droppings containing *T. gondii* oocysts. The tachyzoite and bradyzoite forms of the parasite can persist in water and be accidentally ingested. However, treating water before drinking decreases the risk of *T. gondii* infection.

### Effect of toxoplasmosis on pregnancy outcomes

This study has indicated that *T. gondii* infection was significantly associated with a higher risk of miscarriage among participants. Participants who were seropositive for toxoplasmosis had a threefold increased risk of miscarriages. *T. gondii* infection increases the risk of miscarriage due to its invasion of the placenta, causing inflammation and damage and leading to spontaneous expulsion of the fetuses [12]. This assertion aligns with several studies that linked *T. gondii* infection with spontaneous abortion [87–89]. The study findings highlight the public health significance and call for concerted efforts to integrate *T. gondii* screening with other co-infections to improve pregnancy outcomes.

### Limitation of the study

The sample size of this study was relatively small compared to the study population and may fail to capture the diversity of the population. The serological method was only used to determine the status of *T. gondii* infection without confirmatory tests, such as PCR or avidity tests, with its inherent limitations. The study participants' responses may suffer from recall bias, compromising their reports. This was a health facility-based study and could influence participants' responses. The test results of the co-infections were based on those reported by the facility and were subject to inaccuracies. Notwithstanding these limitations, the outcomes of this study are relevant for public health policy consideration that could improve maternal and neonatal healthcare services.

## 5. Conclusion

The burden of toxoplasmosis and concurrence with HBV and syphilis was high among pregnant women, but they were not at risk for its transmission. Low levels of education, living in rural areas, animals in households, little or no veterinary services, and drinking water from rivers significantly influenced the transmission of *T. gondii*. An infection with *T. gondii* in pregnancy significantly increases the risk of miscarriage. Toxoplasmosis was associated with miscarriage; thus, integrating it into routine antenatal screening could improve pregnancy outcomes. Health promotion interventions such as education on zoonotic risks, improved sanitation, safe water practices, and veterinary care for domestic animals are recommended to reduce infection risk among pregnant women.

## Acknowledgments

We sincerely acknowledge the invaluable contributions of Olivia Ndele, Uginabor Jonathan Matani, Lambon Isaac, Tikpinja Moses, Bright Bindink Kididisil, Beatrice Ellen, and Adam Kofi Jamiru, former Public Health students of the Akenten Appiah-Menka University of Skills Training and Entrepreneurial Development (AAMUSTED), Asante Mampong Campus.

Their dedication to data collection and entry of responses into Google Forms during their voluntary attachment at the health facilities was instrumental to this study. We are also deeply grateful to Mr. Richard Sombeley Assoah and Madam Ndaayaa Margaret Kweasi, the brother and mother of the lead author (Ebenezer Assoah), respectively, for their support throughout the course of this study.

## Author contributions

**Conceptualization:** Ebenezer Assoah, Denis Dekugmen Yar.

**Data curation:** Ebenezer Assoah, Rockson Addy, Joshua Kpieonuma Zineyele.

**Formal analysis:** Ebenezer Assoah, Denis Dekugmen Yar.

**Funding acquisition:** Ebenezer Assoah, Denis Dekugmen Yar.

**Investigation:** Ebenezer Assoah.

**Methodology:** Ebenezer Assoah, Denis Dekugmen Yar.

**Resources:** Ebenezer Assoah.

**Supervision:** Denis Dekugmen Yar, Papa Kofi Amissah-Reynolds.

**Validation:** Denis Dekugmen Yar, Papa Kofi Amissah-Reynolds, Gadafi Iddrisu Balali, Rockson Addy, Joshua Kpieonuma Zineyele.

**Visualization:** Denis Dekugmen Yar, Papa Kofi Amissah-Reynolds, Gadafi Iddrisu Balali, Rockson Addy.

**Writing – original draft:** Ebenezer Assoah.

**Writing – review & editing:** Denis Dekugmen Yar.

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
