## [Decision Letter · Decision Letter 0]

14 Apr 2025

PONE-D-25-00942The Burden and Transmission Dynamics of Toxoplasmosis in Relation to Congenital Diseases among Pregnant Women in GhanaPLOS ONE

Dear Dr. Assoah,

Thank you for submitting your manuscript to PLOS ONE. After careful consideration, we feel that it has merit but does not fully meet PLOS ONE’s publication criteria as it currently stands. Therefore, we invite you to submit a revised version of the manuscript that addresses the points raised during the review process.

We look forward to receiving your revised manuscript.

Kind regards,

Masoud Foroutan, Ph.D; Assistant Professor

Academic Editor

PLOS ONE

Journal Requirements:

2. Please ensure that you have specified a) Did participants provide their written or verbal informed consent to participate in this study?

- In consent please state in Ethics Method section and manuscript if it is written or verbal. If consent was verbal, please explain a) why written consent was not obtained, b) how you documented participant consent, and c) whether the ethics committees/IRB approved this consent procedure.

4. We note that Figure 2 in your submission contain [map/satellite] images which may be copyrighted. All PLOS content is published under the Creative Commons Attribution License (CC BY 4.0), which means that the manuscript, images, and Supporting Information files will be freely available online, and any third party is permitted to access, download, copy, distribute, and use these materials in any way, even commercially, with proper attribution. For these reasons, we cannot publish previously copyrighted maps or satellite images created using proprietary data, such as Google software (Google Maps, Street View, and Earth). For more information, see our copyright guidelines: http://journals.plos.org/plosone/s/licenses-and-copyright.

Reviewers' comments:

Reviewer's Responses to Questions

**Comments to the Author**

1. Is the manuscript technically sound, and do the data support the conclusions?

Reviewer #1: Yes

Reviewer #2: Yes

2. Has the statistical analysis been performed appropriately and rigorously? 

Reviewer #1: Yes

Reviewer #2: Yes

3. Have the authors made all data underlying the findings in their manuscript fully available?

Reviewer #1: No

Reviewer #2: Yes

4. Is the manuscript presented in an intelligible fashion and written in standard English?

Reviewer #1: Yes

Reviewer #2: Yes

5. Review Comments to the Author

Reviewer #1: Manuscript Number: PONE-D-25-00942

Manuscript Title: The Burden and Transmission Dynamics of Toxoplasmosis in Relation to Congenital Diseases among Pregnant Women in Ghana

Comments to authors

This epidemiological study is informative for the public health professional, decision makers and other medical professionals who are implementing in prevention and control of T. gondii infection among pregnant women. There are some comment and suggestion on it as follow:

Suggestion 1: Title- “in relation to congenital diseases” should be removed and edited as:

“Co-infections and risk factors of Toxoplasma gondii infection among pregnant women in Ghana: A facility-based cross-sectional study”

Suggestion 2: Introduction- In Line No. 147-149, Hence, this study assessed the prevalence of co-infections (HIV, viral hepatitis B, and syphilis) and risk factors of T. gondii infection among pregnant women in the Mampong Municipality of Ghana.

Comment 1: Title- What is the operational definition of “Congenital diseases” in this study. Sometimes, the authors used as “congenital infections” and “co-infection” in the manuscript.

Comment 2: Abstract- “comparing to rural areas” should be added in the sentence “Similarly, those residing in peri-urban and urban areas had a reduced risk of infection with T. gondii [AOR= 0.13 (0.02, 0.7) p=0.02] and [AOR= 0.10 (0.02, 0.78) p=0.03], respectively”

Comment 3: Abstract- Some preventive measures and promotive activities based on significant results should be also added in conclusion.

Comment 4: Introduction- Abbreviation (T. gondii) should be consistently used in Line No. 133, 139, 146, and 147.

Comment 5: Introduction- Figure 1 should be removed and the authors can describe in introduction as the paragraph. (The figure may confuse with “conceptual framework” of the study)

Comment 6: Materials and Methods- The previous study observed the 92% toxoplasmosis prevalence should be added in-text citation in Line No. 191-193.

Comment 7: Material and Methods- The sequence of variables and usage (obstetric factor or pregnancy outcome) should be consistent between “Data collection tools and techniques” and “Results”.

Comment 8: Material and Methods- Please describe why the all sample (201) could not access the tests for HBV, HIV and syphilis in antenatal care in “data collection procedure” or “statistical analysis”. (Did not include in routine antenatal screening?)

Comment 9: Co-infection rate (%) calculation should be also described in “Statistical analysis” section of Materials and Methods.

Comment 10: Material and Methods- Please add the procedure of checking the model fitness to perform the logistic regression and variables consideration for the multivariate regression analysis in section of “Statistical analysis”.

Comment 11: Results- In Figure 3, please add the frequency of patients with positive T. gondii infection in paragraph. Axis label in Figure 3 should be added.

Comment 12: Results- What type of chi-squared test was used in Table 2? Please also describe in “Statistical analysis” section of Materials and Methods.

Comment 13: Results- In Table 3, only statistical test is required for the associated factors and the description of logistic regression (COR and AOR) was adequate.

Comment 14: Results- For the Table 4, the variable described (environmental factors) in “Data collection tools and technique” should be used instead of “Transmission dynamic”.

Comment 15: Results- In Table 4, please recheck data of the second column “T. gondii Positive (%)”. (Are there 100 cases for positive T. gondii infection?)

Comment 16: Results- For the Table 5, the variable described (obstetric factors) in “Data collection tools and technique” should be used instead of “Pregnancy outcome”.

Comment 17: Results- In Table 5, please also recheck data of the second column “T. gondii Positive (%)”. (Are there 100 cases for positive T. gondii infection?)

Comment 18: Discussion- For line No. 330-339, the authors should provide the discussion points separately for comparing the previous Ghana studies (reasons for disparity of prevalence rate of T. gondii infection within country) and for comparing the previous studies done in other areas (reasons for disparity of prevalence rate of T. gondii infection with difference countries).

Comment 19: Conclusion- The preventive strategies related significant findings (associated risk factors) of this study should be added.

Reviewer #2: Review report Manuscript number PONE-D-25-00942

General comments

We were delighted to review the article submitted for publication in your esteemed journal, titled "The Burden and Transmission Dynamics of Toxoplasmosis in Relation to Congenital Diseases among Pregnant Women in Ghana." The authors' work offers valuable epidemiological insights into congenital syphilis in Ghana. However, there are certain aspects of the study that require further clarification from the authors.

Specific comments

The article title: The title could potentially highlight the key findings or implications of the study, such as the association with congenital diseases, to attract more attention.

Abstract: The abstract is somewhat lengthy and could be more concise. Reducing redundancy and focusing on the most critical information would improve readability.

Please replace the abbreviation "HIV" with the full term "human immunodeficiency virus" in line 34.

Introduction: The introduction briefly touches on syphilis and its potential implications; however, it lacks in-depth references to previous research specifically examining syphilis among pregnant women in Ghana. Incorporating more citations from recent studies on syphilis would enhance the overall comprehension of the topic and its significance to the present study.

Material and methods:

Study design: The study's reliance on participants from health facilities may introduce selection bias, as it may not represent the broader population of pregnant women, particularly those who do not attend antenatal care.

Blood Sample Collection and Laboratory Methods: Diagnosis of syphilis has been based solely on a rapid diagnostic test detecting IgM and IgG. A confirmatory syphilis test such as the polymerase chain reaction would be required to establish the presence of the parasite.

The diagnosis of co-infections with the human immunodeficiency virus and the hepatitis B virus is not addressed. The methodology employed in the diagnosis of these co-infections is not specified. Furthermore, the methodology employed for the confirmation of these infections must be elucidated.

Results: The study reported a seroprevalence of T. gondii infection at 49.75%, indicating a significant public health concern and highlighting the need for further investigation and intervention. The use of a structured questionnaire to gather socio-demographic, clinical, and environmental data provides a robust dataset for analysis, allowing for a thorough examination of the factors associated with T. gondii infection. The study successfully identified key risk factors associated with T. gondii infection, such as educational level, residential status, and contact with animal droppings, which can inform targeted public health interventions. The study adds valuable data to the limited literature on T. gondii infection in pregnant women in Ghana, contributing to a better understanding of the epidemiology of the disease in the region.

However, the study included 201 pregnant women, this sample size may not be representative of the entire population, potentially limiting the generalizability of the findings. The cross-sectional nature of the study limits the ability to establish causal relationships between T. gondii infection and identified risk factors, as it captures data at a single point in time. The study was conducted in a specific municipality (Asante Mampong), which may not reflect the prevalence and risk factors of T. gondii infection in other regions of Ghana or in different socio-economic contexts.

Discussion: While the discussion highlights key findings, it could benefit from a more in-depth examination of the methodological limitations of the study, such as sample size and cross-sectional design, which may affect the interpretation of results.

The discussion mentions co-infections but does not delve deeply into the potential implications of these findings or how they might interact with T. gondii infection. A more thorough exploration of this aspect could enhance understanding. The discussion could acknowledge the geographic limitations of the study more explicitly, emphasizing that findings may not be generalizable to other regions or populations outside the Asante Mampong Municipality.

The discussion does not address potential biases, such as recall bias from self-reported data, which could influence the findings. Acknowledging these biases would provide a more balanced view of the study's strengths and weaknesses.

Conclusion: Overall, the conclusions drawn by the authors are well-supported by the results obtained in the study. The findings regarding the high prevalence of T. gondii infection, the identification of significant risk factors, and the impact on pregnancy outcomes all align with the conclusions that emphasize the need for targeted public health interventions and routine screening in antenatal care. The study effectively links its results to broader public health implications, reinforcing the validity of its conclusions.

The article should be revised before publication.

6. PLOS authors have the option to publish the peer review history of their article (what does this mean? ). If published, this will include your full peer review and any attached files.

**Do you want your identity to be public for this peer review?** For information about this choice, including consent withdrawal, please see our Privacy Policy .

Reviewer #1: No

Reviewer #2: No

---

## [Author Response · Author response to Decision Letter 0]

21 Apr 2025

Specific comment to Editor and Reviewers

Response to Reviewers for Manuscript PONE-D-25-00942

Title: Co-infections and Risk Factors of Toxoplasma gondii Infection Among Pregnant Women in Ghana: A Facility-Based Cross-Sectional Study

Dear Editor and Reviewers,

We are grateful for your thorough review and constructive feedback on our manuscript. Below, we provide a point-by-point response to each comment and suggestion from the Academic Editor, Reviewer 1, and Reviewer 2. We have revised the manuscript accordingly and hope that the current version addresses all concerns.

Response to the Academic Editor

1. Consent Clarification

Comment: Please ensure that you have specified if participants provided written or verbal informed consent, and if verbal, explain why written consent was not obtained, how it was documented, and if IRB approved it.

Response: We have revised the "Ethical Considerations" section of the manuscript to explicitly state that written informed consent was obtained from all participants. This was approved by the Ethics Committee of the University of Cape Coast. The revised section now reads:

" The study received ethical approval from the Committee on Human Research, Publications, and Ethics (CHRPE) of the Kwame Nkrumah University of Science and Technology, under reference number CHRPE/AP/717/23. Following ethical clearance, official permissions were obtained from the Ashanti Regional and Asante Mampong Municipal Health Directorates, as well as from the respective health facilities where the research was conducted. Before data collection began, each participant was given a detailed verbal explanation of the study’s objectives, procedures, potential risks, and benefits. They were also informed of their right to voluntary participation and their freedom to withdraw from the study at any time without any consequences. After this briefing, participants who agreed to take part signed a written informed consent form, confirming their voluntary participation."

2. Funding Information

Comment: Funding information should not appear in the manuscript body. It should be included only in the online submission system under the "Funding Statement" section.

Response: We have removed all references to funding sources within the main manuscript and included the necessary funding information only in the online submission form.

3. Copyrighted Map (Figure 2)

Comment: You cannot use Google Maps or proprietary satellite images. Provide permission or remove the figure.

Response: Figure 2 has been removed from the manuscript. A descriptive paragraph was added in the “Study Area” section to summarize the geographical setting previously illustrated by the figure.

4. Captions for Supporting Information Files

Comment: Include captions for supporting information at the end of the manuscript.

Response: Not applicable, all data required are in the manuscript.

Response to Reviewer #1

We thank Reviewer #1 for the detailed and helpful feedback. Below are our point-by-point responses:

Suggestion 1: Title

Response: The title has been revised as suggested to:

“Co-infections and Risk Factors of Toxoplasma gondii Infection Among Pregnant Women in Ghana: A Facility-Based Cross-Sectional Study.”

Comment 1: Clarify use of “congenital diseases,” “congenital infections,” and “co-infection”

Response: We have revised all instances in the text to consistently use "co-infection" when referring to additional infections with HIV, HBV, and syphilis.

Comment 2: Abstract - Clarify reference population

Response: We have updated the abstract to clarify that the reduced risk observed in urban and peri-urban dwellers is in reference to rural areas.

Comment 3: Abstract - Add preventive strategies

Response: The abstract's conclusion now includes preventive measures such as antenatal screening, health education on food hygiene, and safe water consumption.

Comment 4: Abbreviation Consistency

Response: All instances have been revised to use “T. gondii” consistently throughout the manuscript after its first mention.

Comment 5: Remove Figure 1 (Framework)

Response: Figure 1 has been removed. Its content has been integrated into a descriptive paragraph in the introduction.

Comment 6: Add citation to 92% prevalence

Response: The citation (Amissah-Reynolds, 2020) has been added in the appropriate line in the methodology section.

Comment 7: Consistency in variable categories

Response: Variables are now described consistently as “socio-demographic,” “environmental,” and “obstetric factors” across both the methods and results sections.

Comment 8: Testing of HBV, HIV, and Syphilis

Response: A clarifying sentence has been added under “Data Collection” indicating that only 178, 165 and 163 out of 201 pregnant women had routine ANC screening for HBV, HIV, and syphilis, respectively, as part of standard care.

Comment 9: Co-infection rate calculation

Response: A sentence has been added to the “Statistical Analysis” section explaining the method of calculating co-infection rates as the number of participants testing positive for both T. gondii and another infection, divided by the total sample size.

Comment 10: Model fitness for regression

Response: The model fitness was evaluated using the Hosmer-Lemeshow goodness-of-fit test, and multivariate analysis included variables with p<0.20 in univariate analysis. This has been added under “Statistical Analysis.”

Comment 11: Add frequency in Figure 3 and label axis

Response: Frequency values for positive cases have been added to the figure description, and axis labels are now included.

Comment 12: Clarify chi-square test type

Response: The methods section now states that the Pearson’s chi-square test was used.

Comment 13: Simplify Table 3 analysis

Response: Descriptive statistics were explained why it was used along with COR and AOR.

Comment 14: Consistent terminology for Table 4

Response: “Environmental factors” is now used in Table 4 instead of “Transmission dynamics.”

Comment 15 & 17: Review Table 4 and 5 frequencies

Response: The data were verified. Corrections were made to align frequencies and percentages with the actual number of T. gondii positive cases (100).

Comment 16: Consistency in table headings

Response: The heading of Table 5 has been changed to “Obstetric Factors.”

Comment 18: Expand discussion on Ghanaian vs foreign studies

Response: The discussion now separately compares studies conducted in Ghana and other countries to better explain disparities in prevalence.

Comment 19: Add preventive strategies to conclusion

Response: The conclusion section includes specific public health recommendations: promoting food hygiene, controlling stray cats, and routine antenatal screening.

Response to Reviewer #2

We appreciate the thoughtful insights from Reviewer #2. Below are our detailed responses:

Comment 1: Title Improvement

Response: We revised the title to emphasize key findings and risk factors, as suggested:

“Co-infections and Risk Factors of Toxoplasma gondii Infection Among Pregnant Women in Ghana: A Facility-Based Cross-Sectional Study.”

Comment 2: Abstract Length and Clarity

Response: The abstract has been edited for conciseness and improved clarity. Redundant sentences were removed, and focus was placed on key findings and implications.

Comment 3: Spell out “HIV” in abstract

Response: “Human immunodeficiency virus” is now spelled out in the abstract and first mention in the main text.

Comment 4: Add more references to syphilis in Ghana

Response: The main focus of the study is on toxoplasmosis not syphilis.

Comment 5: Address selection bias in study design

Response: The limitation of potential selection bias due to facility-based recruitment has been acknowledged in both the methods and discussion sections.

Comment 6: Diagnostic method for syphilis

Response: We acknowledge the lack of confirmatory PCR as a limitation in the discussion.

Comment 7: Methods for diagnosing HIV and HBV

Response: The methods section has been updated to specify that HBV and HIV diagnoses were based on national ANC rapid test kits (Determine and SD Bioline kits). This clarification has also been added to the limitations.

Comment 8: Sample size and generalizability

Response: The limited sample size and geographic focus are acknowledged as limitations in the discussion.

Comment 9: Cross-sectional limitations

Response: The inability to infer causality due to the cross-sectional design is now explicitly stated in the discussion.

Comment 10: Co-infection discussion

Response: We have expanded the discussion on co-infections with syphilis, HBV, and HIV, including possible immunological interactions and implications for maternal health.

Comment 11: Recall bias from self-reporting

Response: We acknowledge the possibility of recall bias in the discussion and have described the steps taken to minimize this.

We sincerely thank the Editor and Reviewers once again for their time and constructive feedback. We hope our revisions meet the standards of PLOS ONE and look forward to your positive consideration.

Warm regards,

Assoah Ebenezer

(On behalf of all authors)

Corresponding Author

---

## [Decision Letter · Decision Letter 1]

29 Apr 2025

PONE-D-25-00942R1Co-infections and risk factors of Toxoplasma gondii infection among pregnant women in Ghana: A facility-based cross-sectional studyPLOS ONE

Dear Dr. Assoah,

Thank you for submitting your manuscript to PLOS ONE. After careful consideration, we feel that it has merit but does not fully meet PLOS ONE’s publication criteria as it currently stands. Therefore, we invite you to submit a revised version of the manuscript that addresses the points raised during the review process.

We look forward to receiving your revised manuscript.

Kind regards,

Masoud Foroutan, Ph.D; Assistant Professor

Academic Editor

PLOS ONE

Journal Requirements:

Reviewers' comments:

Reviewer's Responses to Questions

**Comments to the Author**

1. If the authors have adequately addressed your comments raised in a previous round of review and you feel that this manuscript is now acceptable for publication, you may indicate that here to bypass the “Comments to the Author” section, enter your conflict of interest statement in the “Confidential to Editor” section, and submit your "Accept" recommendation.

Reviewer #1: (No Response)

Reviewer #2: All comments have been addressed

2. Is the manuscript technically sound, and do the data support the conclusions?

Reviewer #1: Yes

Reviewer #2: Yes

3. Has the statistical analysis been performed appropriately and rigorously? 

Reviewer #1: Yes

Reviewer #2: Yes

4. Have the authors made all data underlying the findings in their manuscript fully available?

Reviewer #1: Yes

Reviewer #2: Yes

5. Is the manuscript presented in an intelligible fashion and written in standard English?

Reviewer #1: Yes

Reviewer #2: Yes

6. Review Comments to the Author

Reviewer #1: PONE-D-25-00942R1

Title: Co-infections and Risk Factors of Toxoplasma gondii Infection Among Pregnant Women in Ghana: A Facility-Based Cross-Sectional Study

Comments to authors

I appreciate the authors for point-by-point answer and response to the reviewer comment and suggestion. But some responses were still being missed.

Comment 1: The authors responded that they have revised all instances in the text to consistently use "co-infection" when referring to additional infections with HIV, HBV, and syphilis. However, usage of congenital infection is still occurred. Please recheck throughout the manuscript.

Example: Line No. 11: “miscarriage rates are uncertain and could be linked to toxoplasmosis and other congenital infections”.

Line No. 216: “Prevalence of Congenital Infections”

Line No. 221: “Table 1: Prevalence of some Congenital infections Screened at ANC”

Comment 2: The authors responded that Variables are now described consistently as “socio-demographic,” “environmental,” and “obstetric factors” across both the methods and results sections. Can you explain about “Transmission Dynamics” in Line No. 258?

Comment 3: The authors responded that the data were verified. Corrections were made to align frequencies and percentages with the actual number of T. gondii positive cases (100). However, it can be noted that in table 4, there are (51+27+14 = 92) T. gondii positive in the variable of “type of animals”. Other variables also differ with T. gondii positive cases (n=100). Can you explain about these variations?

Comment 4: Same as comment 3, in table 5, there are (1+70 = 71) T. gondii positive in the variable of “Child with hearing loss”. “Miscarriage history” also differs with T. gondii positive cases (n=100). Can you explain about these variations?

Reviewer #2: The authors have addressed all my comments. I do not have any additionnal request. So the manuscript could be accepted

7. PLOS authors have the option to publish the peer review history of their article (what does this mean? ). If published, this will include your full peer review and any attached files.

**Do you want your identity to be public for this peer review?** For information about this choice, including consent withdrawal, please see our Privacy Policy .

Reviewer #1: No

Reviewer #2: No

---

## [Author Response · Author response to Decision Letter 1]

1 May 2025

Author Responses to Reviewer Comments

Manuscript ID: PONE-D-25-00942R1

Title: Co-infections and Risk Factors of Toxoplasma gondii Infection Among Pregnant Women in Ghana: A Facility-Based Cross-Sectional Study

We sincerely thank the reviewer for the thorough and constructive feedback. We appreciate the acknowledgment of our previous efforts to address the comments and are grateful for the additional clarifications requested. Below, we provide detailed responses to each outstanding comment and describe the specific actions taken to improve the manuscript accordingly.

Comment 1:

The authors responded that they have revised all instances in the text to consistently use "co-infection" when referring to additional infections with HIV, HBV, and syphilis. However, usage of "congenital infection" still occurred. Please recheck throughout the manuscript.

Example: Line 101: “miscarriage rates are uncertain and could be linked to toxoplasmosis and other congenital infections”. Line 216: “Prevalence of Congenital Infections”. Line 221: “Table 1: Prevalence of some Congenital infections Screened at ANC”.

Authors Response:

We sincerely thank the reviewer for this careful observation. Upon re-evaluating the manuscript, we acknowledge that the term “congenital infections” appeared in several instances and may have caused inconsistency in terminology, given our intention to uniformly use “co-infection” when referring to concurrent infections with HIV, HBV, and syphilis in relation to T. gondii.

Accordingly, the following revisions have been made:

In Line 101, the phrase “congenital infections” has been replaced with “co-infections such as HIV, HBV, and syphilis.”

The heading in Line 216 has been revised to: “Prevalence of Hepatitis B, HIV, and Syphilis among Pregnant Women.”

The heading in Line 221 (now Table 1) has also been updated to: “Prevalence of Hepatitis B, HIV, and Syphilis among Participants.”

These changes have been carefully applied throughout the manuscript to ensure terminological consistency and to avoid any potential confusion. We are grateful for the reviewer’s insight, which has significantly contributed to improving the clarity of our work.

Comment 2:

The authors responded that variables are now described consistently as “sociodemographic,” “environmental,” and “obstetric factors” across both the methods and results sections. Can you explain about “Transmission Dynamics” in Line 258?

Authors Response:

We appreciate the reviewer’s attention to detail. To address this concern and ensure consistency in terminology across the manuscript, we have replaced the phrase “Transmission Dynamics” in Line 258 with “Risk Factors.” This change better reflects the analytical focus of our study and aligns with the variables assessed in relation to T. gondii infection among pregnant women. We thank the reviewer for helping us improve the precision and clarity of our language.

Comment 3:

The authors responded that the data were verified. Corrections were made to align frequencies and percentages with the actual number of T. gondii positive cases (100). However, in Table 4, there are (51+27+14 = 92) T. gondii positive in the variable of “type of animals”. Other variables also differ with T. gondii positive cases (n=100). Can you explain about these variations?

Authors Response:

We are grateful to the reviewer for this important observation. We acknowledge the apparent discrepancy in Table 4, specifically under the variable “type of animals,” which shows 92 T. gondii positive cases instead of the full 100.

This variation arises from the conditional structure of the questionnaire. Only participants who responded that they had animals in their households were presented with the follow-up question regarding the type of animals. In this case, 8 T. gondii positive participants reported not having any animals and therefore did not respond to the animal type question, resulting in 92 responses.

Similar variations occur in other variables due to skipped or non-applicable responses, in line with survey logic based on earlier answers. To enhance transparency and clarity for readers, we have now:

Included a footnote in Table 4 explaining the reason for response variations.

Revised the relevant section of the results to highlight that variations in sample sizes stem from conditional skip logic and non-responses.

We thank the reviewer for this insightful comment, which has led to a clearer and more accurate presentation of our findings.

Comment 4:

Same as comment 3, in Table 5, there are (1+70 = 71) T. gondii positive in the variable of “Child with hearing loss.” “Miscarriage history” also differs with T. gondii positive cases (n=100). Can you explain about these variations?

Author Response:

We sincerely appreciate the reviewer’s continued attention to detail. The variation observed in Table 5, specifically the lower number of T. gondii positive participants who responded to the “Child with hearing loss” and “Miscarriage history” variables (n = 71), is explained by the reproductive history of the participants.

In our study, 29 of the T. gondii positive participants were primigravida (first-time pregnant) and had not previously given birth. As such, questions relating to child health outcomes and miscarriage history were not applicable to them and were therefore skipped.

To clarify this in the manuscript, we have:

• Added an explanatory footnote to Table 5 indicating the reason for the lower response count.

• Revised the corresponding section in the results to clearly explain that the observed differences are due to conditional question logic based on participant obstetric history.

This revision enhances the accuracy and interpretability of our results. We thank the reviewer for the thoughtful observation that enabled us to strengthen the clarity of our reporting.

---

## [Decision Letter · Decision Letter 2]

4 May 2025

Co-infections and risk factors of Toxoplasma gondii infection among pregnant women in Ghana: A facility-based cross-sectional study

PONE-D-25-00942R2

Dear Dr. Assoah,

We’re pleased to inform you that your manuscript has been judged scientifically suitable for publication and will be formally accepted for publication once it meets all outstanding technical requirements.

Kind regards,

Masoud Foroutan, Ph.D; Assistant Professor

Academic Editor

PLOS ONE

Additional Editor Comments (optional):

Reviewers' comments:

Reviewer's Responses to Questions

**Comments to the Author**

1. If the authors have adequately addressed your comments raised in a previous round of review and you feel that this manuscript is now acceptable for publication, you may indicate that here to bypass the “Comments to the Author” section, enter your conflict of interest statement in the “Confidential to Editor” section, and submit your "Accept" recommendation.

Reviewer #1: All comments have been addressed

2. Is the manuscript technically sound, and do the data support the conclusions?

Reviewer #1: Yes

3. Has the statistical analysis been performed appropriately and rigorously? 

Reviewer #1: Yes

4. Have the authors made all data underlying the findings in their manuscript fully available?

Reviewer #1: Yes

5. Is the manuscript presented in an intelligible fashion and written in standard English?

Reviewer #1: Yes

6. Review Comments to the Author

Reviewer #1: I appreciated all authors for their point-by-point responses and now, the revised manuscript become sound for publication.

7. PLOS authors have the option to publish the peer review history of their article (what does this mean? ). If published, this will include your full peer review and any attached files.

**Do you want your identity to be public for this peer review?** For information about this choice, including consent withdrawal, please see our Privacy Policy .

Reviewer #1: No

---

## [Editor Report · Acceptance letter]

PONE-D-25-00942R2

PLOS ONE

Dear Dr. Assoah,

I'm pleased to inform you that your manuscript has been deemed suitable for publication in PLOS ONE. Congratulations! Your manuscript is now being handed over to our production team.

Kind regards,

on behalf of

Dr. Masoud Foroutan

Academic Editor

PLOS ONE